# Don't close the book on tocilizumab for the treatment of severe COVID-19 pneumonia–the jury is still out: The Kuwait experience

Yousef Al-Shamali[1,2☉], Yaser M. Ali[3☉], Rawan A. Al-Shamali[4☉]*, Maryam Al-Melahi[5☉], Farah R. Al-Shammari[6☉], Ahmad Alsaber[7☉], Wasl Al-Adsani[8☉]

1 Department of Gastroenterology, Vancouver General Hospital, Vancouver, Canada, 2 Department of Gastroenterology, Jaber Al Sabah Hospital, Kuwait City, Kuwait, 3 Department of Rheumatology, Mubarak Al Kabeer Hospital, Jabriya, Kuwait, 4 Department of Ophthalmology, Kuwait Institute for Medical Specialization, Al Bahar Hospital, Kuwait, 5 Department of Internal Medicine, Amiri Hospital, Kuwait City, Kuwait, 6 Department of Internal Medicine, Jahra Hospital, Kuwait City, Kuwait, 7 Department of Mathematics and Statistics, University of Strathclyde, Glasgow, United Kingdom, 8 Department of Internal Medicine, Department of Infectious Diseases, Mubarak Al-Kabeer Hospital, Kuwait City, Kuwait

☉ These authors contributed equally to this work.
* Rawan.alshamali@hotmail.com

**Data Availability Statement:** All relevant data are within the manuscript.

## Abstract

### Purpose

This cross-sectional observational study aims to report preliminary data from the first experience using tocilizumab for patients with severe acute respiratory syndrome coronavirus-2 (SARS-CoV-2) infection in three of Kuwait's largest public hospitals City.

### Patients and methods

This chart review study examined the benefits of tocilizumab treatment among 127 patients diagnosed with severe coronavirus disease of 2019 (COVID-19) pneumonia.

### Results

90 of 127 patients (71%) survived. Mortality was highest in the elderly with multiple medical conditions.

### Conclusion

Despite the small sample size and retrospective nature of the work, our findings are consistent with recent studies suggesting tocilizumab administration in patients presenting with severe COVID pneumonia with associated hyperinflammatory features conferred mortality benefit.

**Funding:** The authors received no specific funding for this work.

**Competing interests:** The authors declare that they have no known competing financial interests or personal relationships that could have appeared to influence the work reported in this paper.

## Introduction

It has been well over a year since a severe pneumonia-associated respiratory syndrome caused by a new coronavirus 2019 (COVID-19) was identified. Severe acute respiratory syndrome coronavirus 2 (SARS-CoV-2) has become a global public health challenge. The first case of COVID-19 diagnosed in Kuwait was on February 24, 2020. Most patients with COVID-19 exhibit mild to moderate symptoms; however 20% of COVID-19 patients experience severe complications, including acute respiratory distress syndrome (ARDS), and even progress into an intensive care unit (ICU) admission and death [1–3]. Older age, male gender, and chronic medical conditions are now recognized as higher risk for severe disease [3, 4]. A recent report by the world obesity federation revealed higher BMI increases the risk for COVID-19 complications [5]. The rate of obesity and metabolic syndrome in Kuwait ranks among the highest globally [6]. Two earlier studies estimated in-hospital mortality rates for patients admitted to Kuwait public hospitals with severe COVID-19 pneumonia around 39% to 47% [7, 8]. It is therefore no surprise the rate of COVID-19 related hospitalizations in Kuwait is relatively high [9], especially given the association between cardiometabolic conditions and COVID-19 related compications [10, 11].

Early in the pandemic, multiple repurposed drugs were used to treat COVID-19, including hydroxychloroquine, lopinavir-ritonavir, colchicine, azithromycin, among others. All of these medications have since been refuted. One medication that continued to garner much debate is, Tocilizumab [12]. Tocilizumab is a humanized monoclonal antibody against the interleukin-6 receptor. The Federal Drug Agency has approved tocilizumab (FDA) for the treatment of rheumatoid arthritis, systemic juvenile idiopathic arthritis, giant cell arteritis and cytokine release syndrome. It was first used to treat severe COVID-19 pneumonia in China and Italy [13, 14].

This study aims to report preliminary data from the first experience with tocilizumab administered to patients with severe SARS-CoV-2 infection in three of the largest hospitals in Kuwait City. We compared the clinical characteristics of patients who survived vs. deceased patients following treatment with tocilizumab. We also sought to compare the mortality rate before and after tocilizumab's introduction in Kuwait public hospital sector.

## Materials and methods

### Study population and methods

This cross-sectional, multicenter, observational study examined adult patients, 18 years and older, admitted with severe COVID-19 pneumonia from March 1 till May 30, 2020, at three major hospitals in Kuwait City—Jaber Al Ahmed Hospital Mubarak Al-Kabeer Hospital, and Al Jahra Hospital. A chart review of all patients treated with tocilizumab was performed. Inclusion criteria required a diagnosis of severe COVID-19 pneumonia with associated hyperinflammatory features. Severe COVID-19 pneumonia was established based on positive reverse transcription-polymerase chain reaction (RT-PCR) nasopharyngeal swab with associated dyspnea symptoms, respiratory frequency $\geq$ 30/min or blood O2 sat $\leq$93%, plus lung infiltrates on plain chest x-ray imaging. As direct interleukin 6 (IL-6) was not available at the time, this study was undertaken. Inflammatory markers were used as a surrogate measure, either elevated C-reactive protein greater than 100 milligrams per liter (mg/L) or Ferritin greater than 500 microgram per liter (ug/L). Pregnant patients and patients with end-stage liver disease were excluded from the study.

The ethical committee of the Ministry of Health at Kuwait approved the study *1483/2020*. The ethics committee exempted informed consent as this study was a chart review and

retrospective in nature. This study was conducted according to the principles expressed in the Helsinki Declaration.

The information obtained included age, gender, presenting symptoms, type of respiratory support, and comorbid medical conditions. Adverse events following tocilizumab administration were recorded, mainly elevated liver enzyme and secondary infection. Secondary infection was defined as clinical signs of superimposed bacterial pneumonia or bacteremia, based on positive blood, body fluid, or endotracheal samples.

The criteria for tocilizumab administration were uniform across all three hospitals. Tocilizumab was given in the setting of rapidly progressive or insidious respiratory failure (defined peripheral capillary oxygen saturation (SpO2) < 90% on 4 Liters or increasing oxygen requirements over 24 hours) plus elevated C-reactive protein or Ferritin levels. Tocilizumab was administered at a dose of 8 mg/kg (max 800 mg) via intravenous infusion. A second dose was given at the discretion of the treating physician 12–24 hours apart. Other treatments provided to the subjects, included (hydroxychloroquine, Lopinavir-Ritonavir, and corticosteroid), were administered at the treating physician's discretion.

The primary outcome was defined as in-hospital mortality. A secondary outcome was the complication rate following tocilizumab administration, all measured at day 15 post-tocilizumab administration.

Statistical analysis performed using SPSS, version 23.0 (Armonk, NY: IBM Corp). Data were presented as either median (min-max) or, where indicated, the number and percentage. The Wilcoxon signed-rank test was used to compare parameters. A P-value of less than or equal to .05 was considered statistically significant.

## Results

### Demographic characteristics

A total of 127 patients were included in this study. The mean age of the overall patients was 57.1 ± 12.5 years (Table 1). Of the 127 patients, 111 were males (87.4%), and 16 were females (12.6%). The most common symptoms across all the subjects were shortness of breath 97 (76.4%), dry cough 89 (70.1%), fever 81 (63.8%), myalgia 37 (29.1%), nausea 32 (25.2%), and diarrhea 30 (23.6%). All patients required oxygen therapy during their hospital stay, including face mask oxygen in 67 patients (52.8%), high flow nasal cannula in 22 patients (17.3%), non-invasive ventilation in 6 patients (4.7%), and invasive ventilation in 32 patients (25.2%).

### Clinical outcome

On day 15 of follow-up, 90 patients were alive (71.9%), and 37 patients died (29.1%], of which 41 (45.5%) were discharged from the hospital. The mean age of the alive group was 54.3 years relative to the deceased group's average age of 64.1 years. More diabetic patients were observed in the deceased group 23 (62.2%) vs. the alive group 38 (42.2%). The deceased group also had significantly more patients with comorbid chronic medical conditions, including diabetes mellitus, hypertension, coronary artery disease, and chronic kidney disease (Table 2).

### Adverse events

Adverse events following the administration of tocilizumab were monitored for up to 15 days. The alive group had a significantly lower secondary infection rate than the deceased group (6.7% vs. 37.8%). One major sentinel event occurred in a deceased patient attributed to tocilizumab (Table 3).

**Table 1. Demographic characteristics, most common presenting symptom(s), and type of respiratory support.**

| Characteristics | Value |
|---|---|
| Mean age (range)–years | 57.1 ± 12.5 |
| Gender | |
| Males | 111 (87.4%) |
| Females | 16 (12.6%) |
| Common comorbid medical Conditions | |
| Diabetes Mellitus | 61 (48.0%) |
| Hypertension | 58 (45.7%) |
| Coronary Artery Disease | 22 (17.3%) |
| Chronic Kidney Disease | 7 (5.51%) |
| Most Common Presenting Symptom(s) | |
| Shortness of breath | 97 (76.4%) |
| Dry cough | 89 (70.1%) |
| Fever | 81 (63.8%) |
| Myalgia | 37 (29.1%) |
| Nausea | 32 (25.2%) |
| Diarrhea | 30 (23.6%) |
| Type of Respiratory Support | |
| Face mask oxygen | 67 (52.8%) |
| High flow nasal cannula | 22 (17.3%) |
| Noninvasive ventilation | 6 (4.7%) |
| Invasive (mechanical) ventilation | 32 (25.2%). |

## Discussion

In some patients with severe COVID-19 pneumonia, IL-6 seems to play an essential role in the pathogenesis of cytokines release syndrome (CRS). A recent study has shown a significant elevation of IL-6 in some COVID-19 patients, especially in critically ill patients [15]. Elevated levels of IL-6 in the blood have been reported to be predictive of a fatal outcome in patients with COVID-19 [16]. Tocilizumab, a recombinant human IL-6 monoclonal antibody, binds to soluble and membrane-bound IL-6 receptors, thus blocking IL-6 signaling. It is speculated that the blocking of the IL-6 signal, in turn, will arrest the inflammatory process in patients with CRS.

Multiple observational studies have demonstrated the efficacy of tocilizumab in curtailing CRS in patients with severe COVID-19 pneumonia. Tocilizumab appeared to offer benefits in reducing inflammation, oxygen requirements, vasopressor support, and mortality [17–23]. However, most of the studies favoring tocilizumab's use were hampered by poor design and small sample size.

**Table 2. Clinical outcome.**

| | Alive group N = 90 | Deceased group N = 37 | P-value |
|---|---|---|---|
| Age (mean) | 54.3 | 64.1 | <0.001 |
| Males (%) | 81 (72.8%) | 30 (27.2%) | 0.19 |
| Female (%) | 9 (56.2%) | 7 (43.8%) | <0.41 |
| Diabetes Mellitus | 38 (42.2%) | 23 (62.2%) | 0.04 |
| Hypertension | 36 (40.0%) | 22 (59.5%) | 0.05 |
| Coronary Artery Disease | 10 (11.1%) | 12 (32.4%) | 0.007 |
| Chronic Kidney Disease | 2 (2.22%) | 5 (13.5%) | 0.02 |

**Table 3. Prevalence of adverse events post tocilizumab administration.**

|  | Alive group N = 90 | Deceased group N = 37 | P-value |
|---|---|---|---|
| Elevated Liver enzymes | 9 (10.0%) | 3 (8.11%) | 1.000 |
| Secondary Infection | 6 (6.67%) | 14 (37.8%) | <0.001 |
| Perforated Viscus | 0 (0.00%) | 1 (2.70%) | 0.291 |

Somers and colleagues performed a retrospective, real-world design, which included mechanically ventilated patients with COVID-19 who did (N = 78) and did not (N = 76) receive a single dose of 8 mg/kg tocilizumab [24]. Tocilizumab was associated with a 45% mortality reduction. Recent randomized controlled studies have shown mixed results concerning the efficacy of tocilizumab. In the CORIMUNO-TOCI-1 trial, a randomized, open-label study in Italy, there was no difference in clinical outcome between the tocilizumab group compared to usual care group (27% vs. 28%) in moderate or severe COVID-19 pneumonia [25]. In the RCT-TCZ COVID-19 study group, an open-label trial from France, the primary outcome, mortality, and need for increase ventilation, favored tocilizumab compared to the usual care group (24% vs. 36%). However, by day 28, there was no mortality difference between the two groups [26]. In the STOP-COVID trial, a large trial from the US involving roughly 430 critically ill COVID-19 patients who received tocilizumab upon ICU, findings showed mortality benefits associated with the tocilizumab group nearing a 30% reduction in mortality risk [27]. In the BACC Bay tocilizumab trial, there was no difference between the tocilizumab group versus the standard of care for the need for mechanical ventilation or death over a four-week period [28].

Two industry-sponsored double-blind, randomized controlled trials, COVACTA and EMPACTA, showed mixed results in the use of tocilizumab in hospitalized patients with severe COVID-19 pneumonia [29, 30]. In the COVACTA trial, tocilizumab did not improve clinical status or reduce mortality at day 28. However, time to hospital discharge and duration of intensive care unit (ICU) stay both appeared to be shorter in the tocilizumab group. In the EMPACTA trial, tocilizumab reduced progression toward mechanical ventilation but did not affect mortality.

More recently, two large international trials demonstrated mortality benefits and reduced disease burden with the use of tocilizumab. In the REMAP-CAP study [31], compared with standard care, tocilizumab was associated with more days free of organ support and lower in-hospital mortality. In the RECOVERY trial [32], 2022 patients were randomly allocated to receive tocilizumab and 2094 patients who had usual care. Findings showed 596 of the patients (29%) in the tocilizumab group died within 28 days compared with 694 (33%) patients in the usual care group (rate ratio 0·86; 95% confidence interval [CI] 0·77–0·96; p = 0·007). Tocilizumab also increased the probability of discharge alive within 28 days from 47% to 54% (rate ratio 1.23, 95% CI 1.12 to 1.34; P<0.0001). Treatment with tocilizumab produced a five-day reduction in hospital stays when used in addition to standard care, significantly reducing the chance of progressing to invasive mechanical ventilation or death from 3% to 33% (risk ratio 0.85; 95% CI 0.788 to 0.93, P = 0.0005).

This study's main finding was that on day 15, following tocilizumab treatment, approximately 71% of the patients were either stable or discharged from the hospital. The mortality rate in our study was 29%. In contrast, mortality rates for patients classified as having severe pneumonia in Kuwait before introducing tocilizumab into clinical practice were estimated at around 39% to 47% [8, 9]. Our findings are consistent with previous studies [33–35], namely older males with chronic medical conditions, including diabetes mellitus, hypertension, coronary artery disease, and chronic renal failure, had a higher risk of death from COVID-19.

The main concern regarding the use of tocilizumab therapy is the occurrence of severe infections [36]. In this study, 14 patients (37.8%) experienced secondary bacterial infections after tocilizumab treatment in the deceased and six patients (6.7%) in the alive group. Additionally, one sentinel event occurred in one patient in the deceased group; bowel perforation was attributed to tocilizumab use.

Our study has several limitations. The observational nature and the small size precluded definitive conclusions on tocilizumab's efficacy in patients with severe COVID-19 pneumonia. A confounding variable that not controlled for in this study was other treatment modalities, particularly corticosteroid. It is unclear whether surrogate markers for interleukin 6 (IL-6) may have influenced the management and, ultimately, this study's results. The sequential nature of the comparison group rendered any conclusion indefinite. Finally, our sample consisted mainly of males. It is not clear whether a more balanced sample would have influenced the results.

## Conclusion

In this small multicenter observational study, we found a trend toward a mortality benefit favoring the use of tocilizumab in the treatment of severe COVID-19 pneumonia complicated by hyperinflammatory immune response. Our findings are consistent with the recently updated guidelines from the National Institute of Health [37] and Infectious Diseases Society of America [38] regarding the use of tocilizumab in patients with severe COVID-19 pneumonia with elevated systemic inflammatory markers.

## Author Contributions

**Conceptualization:** Yousef Al-Shamali.

**Data curation:** Rawan A. Al-Shamali, Maryam Al-Melahi, Farah R. Al-Shammari.

**Formal analysis:** Yaser M. Ali, Ahmad Alsaber.

**Methodology:** Yousef Al-Shamali.

**Writing – original draft:** Yousef Al-Shamali.

**Writing – review & editing:** Yousef Al-Shamali, Maryam Al-Melahi, Wasl Al-Adsani.

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
