## [Decision Letter · Decision Letter 0]

19 Mar 2021

PONE-D-20-38350

Don't Close the Book on Tocilizumab for the Treatment of Severe COVID-19 Pneumonia – The Jury is still out.  The Kuwait Experience

PLOS ONE

Dear Dr. alshemali,

Thank you for submitting your manuscript to PLOS ONE. After careful consideration, we feel that it has merit but does not fully meet PLOS ONE’s publication criteria as it currently stands. Therefore, we invite you to submit a revised version of the manuscript that addresses the points raised during the review process.

We look forward to receiving your revised manuscript.

Kind regards,

Aleksandar R. Zivkovic

Academic Editor

PLOS ONE

2. Thank you for stating in the text of your manuscript "The ethical committee of the Ministry of Health at Kuwait approved the study 1483/2020. The ethics committee exempted informed consent as this study was a chart review and retrospective in nature. All investigations were conducted according to the principles expressed in the Helsinki Declaration."

Please also add this information to your ethics statement in the online submission form.

3. Please include the date(s) on which you accessed the databases or records to obtain the retrospective data used in your study.

'The authors declare that they have no known financial interests to disclose that could have appeared to influence the work reported in this paper. '

Reviewer's Responses to Questions

**Comments to the Author**

1. Is the manuscript technically sound, and do the data support the conclusions?

Reviewer #1: Partly

Reviewer #2: Yes

Reviewer #3: Yes

2. Has the statistical analysis been performed appropriately and rigorously? 

Reviewer #1: Yes

Reviewer #2: No

Reviewer #3: No

3. Have the authors made all data underlying the findings in their manuscript fully available?

Reviewer #1: Yes

Reviewer #2: No

Reviewer #3: Yes

4. Is the manuscript presented in an intelligible fashion and written in standard English?

Reviewer #1: No

Reviewer #2: Yes

Reviewer #3: Yes

5. Review Comments to the Author

Reviewer #1: The data presented in this study is very helpful. It is a great addition to the portfolio of tocilizumab use for COVID-19.

Below are a few suggestions:

1. Title:

It would be better if the title follows a uniform pattern (title case). Below is a suggestion:

"Don't Close the Book on Tocilizumab for the Treatment of Severe COVID-19 Pneumonia – the Jury Is Still Out: The Kuwait Experience."

2. Abbreviations in the abstract and within the text:

- Add both the spelled-out version and the short form for SARS-CoV-2 and COVID-19 after one another in the abstract instead of having the abbreviations only

- Follow the same pattern in the introduction the first time these abbreviations are used (COVID-19, FDA, RT-PCR, etc.)

- Use the same naming pattern for all cytokines (full name and acronym when mentioning the name for the first time, and then use either the full name of the acronym, but not both or the acronym that was not listed next to the full name first)

- The same applies for using TCZ in the conclusion although the abbreviation was not mentioned before

3. Upper case:

There is no need to write tocilizumab and other medications, biomarkers, and other terms (such as liters) in upper case within the text.

4. Discussion:

Comparing the mortality rate with previous studies is a great addition, but it would be better if the treatments of these studies are compared with the treatment given to this study subjects. If it is the same (hydroxychloroquine, Lopinavir-Ritonavir, and corticosteroid), then this should be mentioned too.

5. Conclusion:

- Write Parr et al. instead of Parr

6. English Language:

The article is well-structured, but it would be a good idea to seek independent editorial help to make the language clearer, error-free, and consistent.

Reviewer #2: In this study authors were reported the tocilizumab administration in patients with severe COVID-19. Guaraldi et al (https://doi.org/10.1016/S2665-9913(20)30173-9), Hernández-Mora et el (https://doi.org/10.1016/j.ijid.2020.10.045), Colaneri et al (10.3390/microorganisms8050695) and several previous studies reported similar findings.

he study is short for a original article (Short communication is suggested)

The title should be change to a more informative and the study. First paragraph of introduction is too general. 87.4% of the study population are male and it makes a concern in the results.

Reviewer #3: PLOS ONE Review

Article title: Don't Close the Book on Tocilizumab for the Treatment of Severe

COVID-19 Pneumonia – The Jury is still out. The Kuwait Experience

Several concerns regarding the manuscript are described below:

Method

1. The authors used "Wilcoxon signed‐rank test" to analyze the data. This test is designed for a paired group, such as pre-and post-treatment. But when we look up Table 2, this analysis seems not appropriate.

Results

1. The author mentioned, "The mean age of the alive group was 54.3 years relative to the deceased group average age of 64.1 years." However, it should be noted that the deceased group was significantly older than the alive group. This should be explained as one of the possible factors for mortality and may be associated with chronic illnesses such as DM and hypertension.

2. For age, it would be better to include standard deviation in Table 2.

3. For the outcome, is there data regarding the improvement of respiratory support after TCZ treatment?

4. Table 3. When there is no evidence of neutropenia, rash, anaphylaxis, why the percentage in table 3 were 100%?

Discussion

1. The authors mentioned, "Consistent with previous studies, the deceased group were predominantly older males who tended to have…..". However, there is no explanation or data to support this statement in the results section, and all of a sudden, it was described in the discussion.

2. Since TCZ is an immunosuppressive agent, it is not surprising that secondary infection is observed in both groups. Still, I think the authors need to add an additional explanation for this matter.

6. PLOS authors have the option to publish the peer review history of their article (what does this mean?). If published, this will include your full peer review and any attached files.

Reviewer #1: No

Reviewer #2: **Yes: **Masoud Nouri-Vaskeh

Reviewer #3: No

---

## [Author Response · Author response to Decision Letter 0]

20 Jun 2021

Thank you for giving us the opportunity to submit a revised draft of the manuscript “Don't Close the Book on Tocilizumab for the Treatment of Severe COVID-19 Pneumonia – the Jury Is Still Out: The Kuwait Experience” Cognitive Psychology of Metrical Constructs” for publication.

We appreciate the time and effort that you and the reviewers dedicated to providing feedback on our manuscript and are grateful for the insightful comments on and valuable improvements to our paper.

We have incorporated most of the suggestions made by the reviewers. Those changes are highlighted within the manuscript. 

Reviewer #1: The data presented in this study is very helpful. It is a great addition to the portfolio of tocilizumab use for COVID-19.

Below are a few suggestions:

1. Title:

It would be better if the title follows a uniform pattern (title case). Below is a suggestion:

"Don't Close the Book on Tocilizumab for the Treatment of Severe COVID-19 Pneumonia – the Jury Is Still Out: The Kuwait Experience."

Author response: Thank you!

2. Abbreviations in the abstract and within the text:

- Add both the spelled-out version and the short form for SARS-CoV-2 and COVID-19 after one another in the abstract instead of having the abbreviations only

- Follow the same pattern in the introduction the first time these abbreviations are used (COVID-19, FDA, RT-PCR, etc.)

- Use the same naming pattern for all cytokines (full name and acronym when mentioning the name for the first time, and then use either the full name of the acronym, but not both or the acronym that was not listed next to the full name first)

- The same applies for using TCZ in the conclusion although the abbreviation was not mentioned before

Author response: Accordingly, throughout the manuscript, we have revised and edited all abbreviations and capitalizations. 

3. Upper case:

There is no need to write tocilizumab and other medications, biomarkers, and other terms (such as liters) in upper case within the text.

Author response: Thank you! This has been addressed. 

4. Discussion:

Comparing the mortality rate with previous studies is a great addition, but it would be better if the treatments of these studies are compared with the treatment given to this study subjects. If it is the same (hydroxychloroquine, Lopinavir-Ritonavir, and corticosteroid), then this should be mentioned too.

Author response: We agree with the reviewers, accordingly we have edited our conclusion and introduction to reflect such.

5. Conclusion:

- Write Parr et al. instead of Parr

6. English Language:

The article is well-structured, but it would be a good idea to seek independent editorial help to make the language clearer, error-free, and consistent.

Author response: Thank you, we have done our best to follow these comments.

Reviewer #2: In this study authors were reported the tocilizumab administration in patients with severe COVID-19. Guaraldi et al (https://doi.org/10.1016/S2665-9913(20)30173-9), Hernández-Mora et el (https://doi.org/10.1016/j.ijid.2020.10.045), Colaneri et al (10.3390/microorganisms8050695) and several previous studies reported similar findings.

he study is short for a original article (Short communication is suggested)

The title should be change to a more informative and the study. First paragraph of introduction is too general. 87.4% of the study population are male and it makes a concern in the results.

 Author response: Thank you, the introduction has been edited in an effort to be more concise and informative, new studies added includung RECOVERY, and REMAP-CAP trial.

Reviewer #3: PLOS ONE Review

Article title: Don't Close the Book on Tocilizumab for the Treatment of Severe

COVID-19 Pneumonia – The Jury is still out. The Kuwait Experience

Several concerns regarding the manuscript are described below:

Method

1. The authors used "Wilcoxon signed?rank test" to analyze the data. This test is designed for a paired group, such as pre-and post-treatment. But when we look up Table 2, this analysis seems not appropriate.

Author response: We have revised our statical analysis, and edited table 2 p values to reflect this. Thank you

Results

1. The author mentioned, "The mean age of the alive group was 54.3 years relative to the deceased group average age of 64.1 years." However, it should be noted that the deceased group was significantly older than the alive group. This should be explained as one of the possible factors for mortality and may be associated with chronic illnesses such as DM and hypertension.

Author response: Thank you for pointing this out. The reviewer is correct, and we have addressed this in the conclusion. 

2. For age, it would be better to include standard deviation in Table 2.

3. For the outcome, is there data regarding the improvement of respiratory support after TCZ treatment?

4. Table 3. When there is no evidence of neutropenia, rash, anaphylaxis, why the percentage in table 3 were 100%?

Author response: Thank you for pointing this out. We have edited Table 3 to reflect the correct results, and omitted neutropenia, rash, anaphylaxis.

Discussion

1. The authors mentioned, "Consistent with previous studies, the deceased group were predominantly older males who tended to have…..". However, there is no explanation or data to support this statement in the results section, and all of a sudden, it was described in the discussion.

2. Since TCZ is an immunosuppressive agent, it is not surprising that secondary infection is observed in both groups. Still, I think the authors need to add an additional explanation for this matter.

Author response: Thank you for your feedback, we have attempted to reconstruct the conclusion to reflect your comments.

---

## [Editor Report · Decision Letter 1]

25 Jun 2021

Don't Close the Book on Tocilizumab for the Treatment of Severe COVID-19 Pneumonia – The Jury is still out.  The Kuwait Experience

PONE-D-20-38350R1

Dear Dr. alshemali,

We’re pleased to inform you that your manuscript has been judged scientifically suitable for publication and will be formally accepted for publication once it meets all outstanding technical requirements.

Kind regards,

Aleksandar R. Zivkovic

Academic Editor

PLOS ONE
---

## [Editor Report · Acceptance letter]

12 Aug 2021

PONE-D-20-38350R1 

Don't close the book on Tocilizumab for the treatment of severe COVID-19 Pneumonia – the jury is still out: The Kuwait Experience 

Dear Dr. Al-Shamali:

I'm pleased to inform you that your manuscript has been deemed suitable for publication in PLOS ONE. Congratulations! Your manuscript is now with our production department. 

Kind regards, 

on behalf of

Dr. Aleksandar R. Zivkovic 

Academic Editor

PLOS ONE